# Evaluating the Absolute Calibration Accuracy and Stability of AIRS Using the CMC SST

**Hartmut H. Aumann \*, Steven E. Broberg, Evan M. Manning[iD], Thomas S. Pagano and Robert C. Wilson**

Jet Propulsion Laboratory, California Institute of Technology, Pasadena, CA 91101, USA; Steven.E.Broberg@jpl.nasa.gov (S.E.B.); Evan.M.Manning@jpl.nasa.gov (E.M.M.); Thomas.S.Pagano@jpl.nasa.gov (T.S.P.); Robert.C.Wilson@jpl.nasa.gov (R.C.W.)

\* Correspondence: Hartmut.Aumann@jpl.nasa.gov

**Abstract:** We compare the daily mean and standard deviation of the difference between the sea surface skin temperature (SST) derived from clear sky Atmospheric InfraRed Sounder (AIRS) data from seven atmospheric window channels between 2002 and 2020 and collocated Canadian Meteorological Centre (CMC) SST data from the tropical oceans. After correcting the mean difference for cloud contamination and diurnal effects, the remaining bias relative to the CMC SST, is reasonably consistent with estimates of the AIRS absolute accuracy based on the uncertainty of the pre-launch calibration. The time series of the bias produces trends well below the 10 mK/yr level required for climate change evaluations. The trends are in the 2 mK/yr range for the five window channels between 790 and 1231 cm$^{-1}$, and +5 mK/yr for the shortwave channels. Between 2002 and 2020, the time series of the standard deviation of the difference between the AIRS SST and the CMC SST dropped fairly steadily to below 0.4 K in several AIRS window channels, a level previously only seen in gridded SST products relative to the Argo buoys.

**Keywords:** Infrared; hyperspectral; climate

---

## 1. Introduction

The absolute radiometric calibration accuracy of any sounder is a complicated function of its design, its on-orbit thermal environment, likely degradation on orbit, and the scene temperature. The new generation of hyperspectral infrared sounders was designed to produce very accurate and stable data, to meet the 100 mK absolute accuracy and 10 mK/yr stability required for "climate quality" [1]. The achieved absolute accuracy and stability of the sounders at this level is uncertain. Pagano, T.S. et al. [2] estimated the Atmospheric InfraRed Sounder (AIRS) absolute calibration uncertainty based on the SI traceability of the pre-launch calibration to be in the 50 to 200 mK range for 300K scenes. Lower bounds on the absolute accuracy can be established by noting that most sounders make measurements in channels which have identical or functionally identical spectral response functions. The relative brightness temperature differences measured with these channels should show no bias. IASI (Infrared Atmospheric Sounder Interferometer) [3] on the on the MetOp satellites has four detectors, the Crosstrack Interferometer Sounder, CrIS [4] on SNPP and JPSS1 satellites has nine detectors in each of three bands at nominally identical frequencies. The Atmospheric Infrared Sounder, AIRS [5], has several channels with nearly identical spectral response functions. Differences between these nominally identical channels and trends in the differences constitute lower bounds on the absolute accuracy and stability of the calibration. They are lower bounds because there are shared elements in the calibration, which cancel in the difference. Another method is to note that all hyperspectral sounders make measurements in many atmospheric window channels. Each of these

window channels can be used to derive a surface temperature. The differences between the derived temperatures and a reliable surface truth can be used as a measure of the absolute radiometric accuracy and stability.

The general approach of comparing a sea surface skin temperature (SST) derived directly from AIRS clear sky Level 1B radiances to a gridded SST product was proposed before launch and was subsequently tested, most recently in Aumann et al., 2019, using the NOAA generated RTG (Real Time Global) SST [6]. The RTGSST became noisy in about 2017 [7], and was discontinued in 2019. Deriving an SST directly from individual window channels under clear sky conditions retains traceability to the calibration. Note that the Level 2 product, which uses the best 350 of 2378 AIRS channels to simultaneously derive a surface skin temperature and temperature and water vapor profiles from cloud-cleared and tuned level 1B radiances [8] loses the traceability of the calibration of individual channels.

For the ground truth we use the Canadian Meteorological Centre (CMC) SST, which has been produced on a 0.2 degree grid since 1991 (v2.0) and on a 0.1 degree grid since 2016 (v3.0). In the following, we refer to the CMC SST simply as CMC. The primary references for the CMC are in situ observations of the SST by buoys (excluding Argo) and ships from the International Comprehensive Ocean-Atmosphere Data Set (ICOADS) program. The grid is filled by optimally interpolating between the buoy measurements with surface temperatures deduced from space-borne sensors and ship reports. Sensors used in the production of the CMC include the AVHRR from NOAA-18 and 19, the European Meteorological Operational-A (METOP-A) and Operational-B (METOP-B), and data from the Advanced Microwave Scanning Radiometer 2 (AMSR2) onboard the GCOM-W satellite. No AIRS data are used for the CMC production. In a decadal average, the CMC agrees with the independent Argo buoys at the 10 mK level and has a trend of −1.9 ± 1.0 mK/yr (2 sigma) [9].

## 2. Data

We used the L1B v5 calibrated AIRS data available from the GSFC/DISC since September 2002. AIRS on the NASA's Earth Observing System (EOS) Aqua spacecraft is in a 1:30 AM ascending node polar orbit at 703 km altitude. We derived seven independent SSTs for seven independent window channels at 2615, 2508, 1231, 1128, 961, 901 and 790 cm$^{-1}$, representing seven of the fifteen AIRS focal plane detector modules under clear sky conditions for the 30S-30N oceans. Table 1 summarizes the associate module names and absolute calibration uncertainty for 290K scenes based on Pagano et al. 2020.

**Table 1.** The seven Atmospheric InfraRed Sounder (AIRS) window channels and associated module ID, atmospheric correction, and calibration uncertainty.

| Channel [cm$^{-1}$] | Detector Module | Typical atm. Correction [K] | Calibration Uncertainty at 290K [K] |
|---|---|---|---|
| 2615.3 | M1a | 1.0 | 0.11 |
| 2508.1 | M2a | 2.2 | 0.05 |
| 1231.3 | M4d | 2.9 | 0.08 |
| 1128.5 | M5 | 2.7 | 0.22 |
| 961.4 | M7 | 2.1 | 0.05 |
| 901.0 | M8 | 2.7 | 0.12 |
| 790.3 | M9 | 4.7 | 0.16 |

We use a clear sky filter based on a 3 × 3 footprint spatial coherence test (SCT [7]), which basically measures the absolute value of the difference between the brightness temperature of the center pixel at 1231 cm$^{-1}$ and its four nearest neighbors. Samples where this difference is less than a 0.5 K threshold and where the calculated SST using the 1231 cm$^{-1}$ channel differs from the CMC by less than 4 K are

identified as SCT clear. The latter condition eliminates low stratus clouds, akin to a 10 sigma rejection test. The absolute calibration cancels in the SCT, since the same 1231 cm$^{-1}$ detector is used to make the measurements of the nearest neighbors. Since the noise-equivalent delta temperature of this channel is 0.07 K, the impact of random noise on the SCT using a 0.5 K threshold is negligible. Our analysis uses the daily mean and standard deviation (stddev) of the observed brightness temperature (obs) and the brightness temperature calculated (calc). Although the selected channels are in atmospheric windows, atmospheric water vapor causes absorption ranging from 1 K at 2615 cm$^{-1}$ to 4.7 K at 790 cm$^{-1}$ (Table 1, column 3).

Defining bt$_{airs}$ as the observed brightness temperature in a window channel, tr as the combined corrections for atmospheric transmission and surface emissivity [10], then the relationship between obs, calc, CMC and SST$_{airs}$ is shown in Equation (1).

$$(obs\text{-}calc) = bt_{airs} - (CMC\text{-}tr) = (bt_{airs} + tr)\text{-}CMC = SST_{airs}\text{-}CMC. \tag{1}$$

The difference between the 1231.3 and 1227.7 cm$^{-1}$ channels is used to derive the atmospheric transmission correction due to water vapor [7] for all channels. Defining Q = bt1231 − bt1227, we can write

$$SST_{airs} = bt_{airs} + a_0 + a_1 * Q + a_2 * Q^2 + a_3/\cos(sza) \tag{2}$$

where sza is the satellite zenith angle ( between −50 and +50 degrees). The coefficients were regression trained on 1403 open ocean profiles from the European Centre for Medium-range Weather Forecasting (ECMWF) as described in Aumann et al. 2019. The ocean profiles were converted to the AIRS spectra training set using SARTA, the Stand Alone Radiative Transfer Model (RTM) developed for AIRS [11]. SARTA is based on the 2008 version of HITRAN with a pre-release of version 3.2 of the MT_CKD water continuum.

For each day we matched the longitudes and latitudes of the clear ocean footprints within ±30 degrees of the equator to the nearest grid point of the CMC. We calculated SST$_{airs}$ for each clear footprint and evaluated the daily mean and standard deviation of SST$_{airs}$-CMC. The daily number of clear SST matchups, fluctuates daily and seasonally, but is typically about 10,000.

## 3. Results

Figures 1–5 show the time series of the mean and the standard deviation for the day and night overpasses for the midwave channels. The night mean has been shifted by 0.38 K to make the day and night plots approximately overlay. Figure 6 shows the results from night overpasses for the 2615 and 2508 cm$^{-1}$ "shortwave" channels. Daytime results from the shortwave band cannot be used because of reflected solar light. Results from all channels are presented in Table 2.

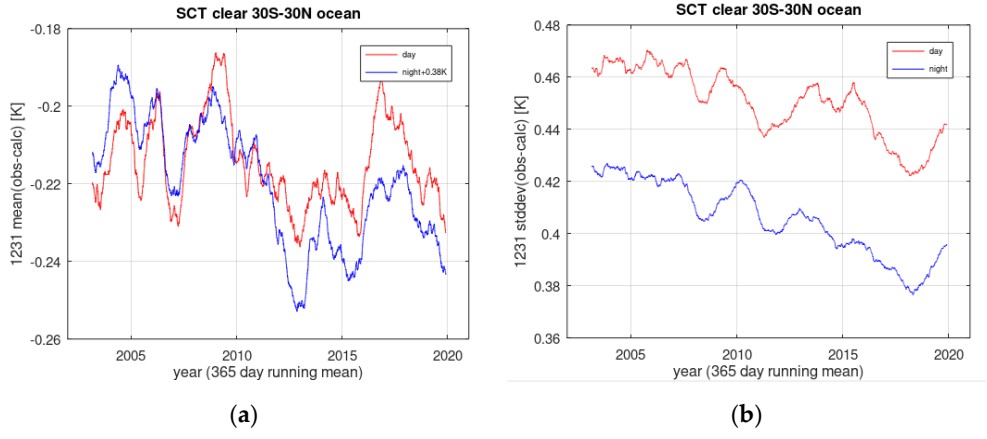

**Figure 1.** (**a**) Mean and (**b**) standard deviation at 1231 cm$^{-1}$ with one-year smoothing.

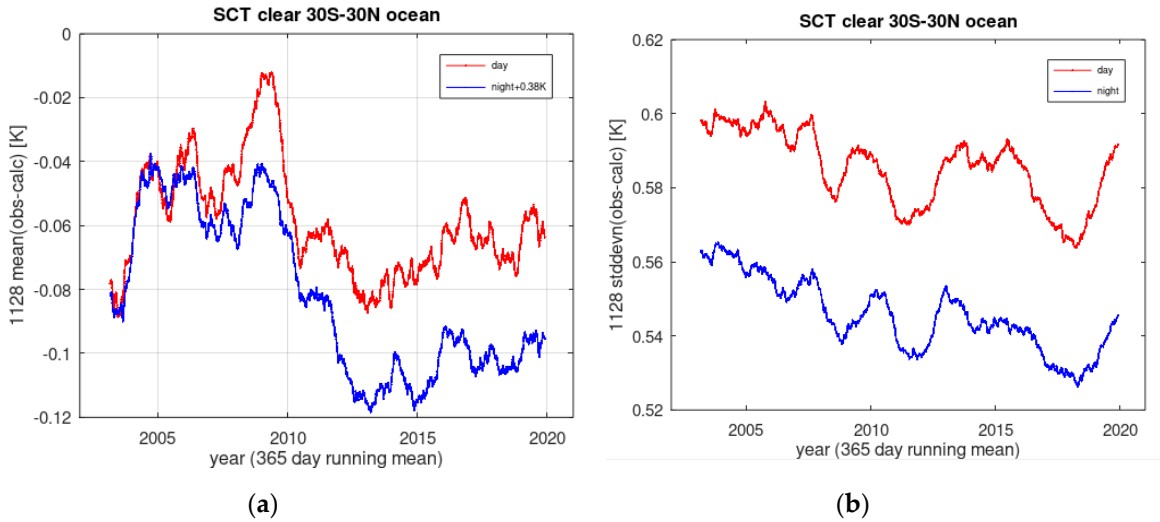

**Figure 2.** (**a**) Mean and (**b**) stddev at 1128 cm$^{-1}$ with one-year smoothing.

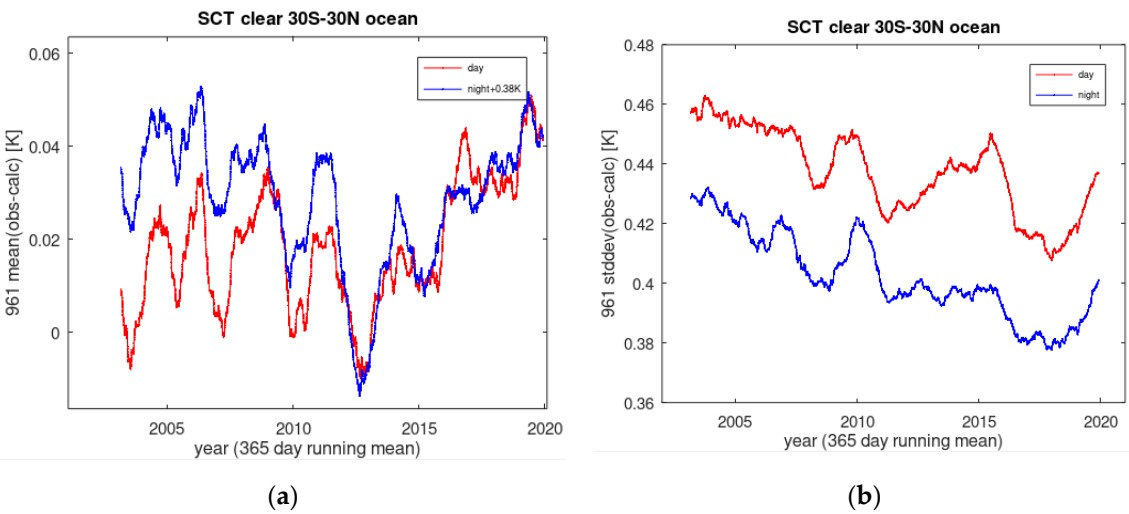

**Figure 3.** (**a**) Mean and (**b**) standard deviation at 961 cm$^{-1}$ with one-year smoothing.

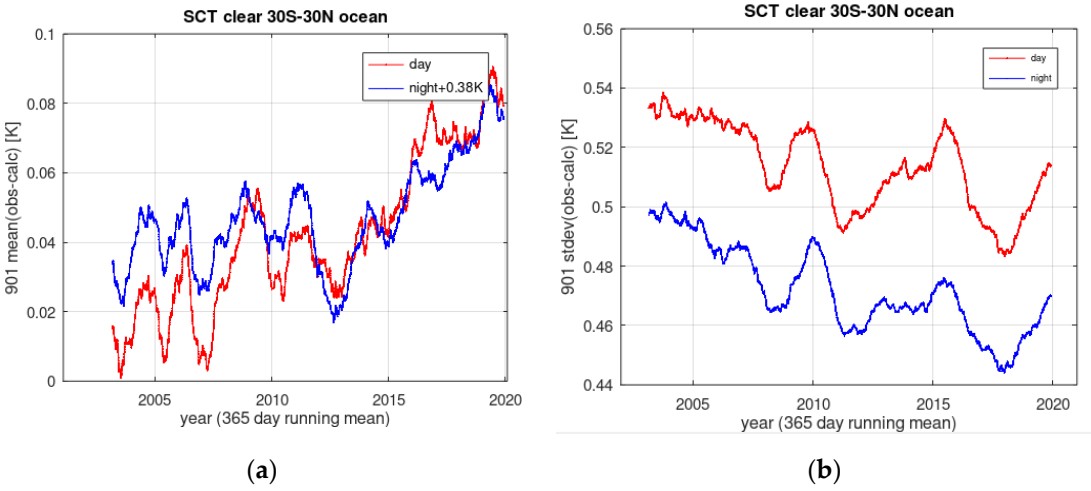

**Figure 4.** (**a**) Mean and (**b**) standard deviation at 901 cm$^{-1}$ with one-year smoothing.

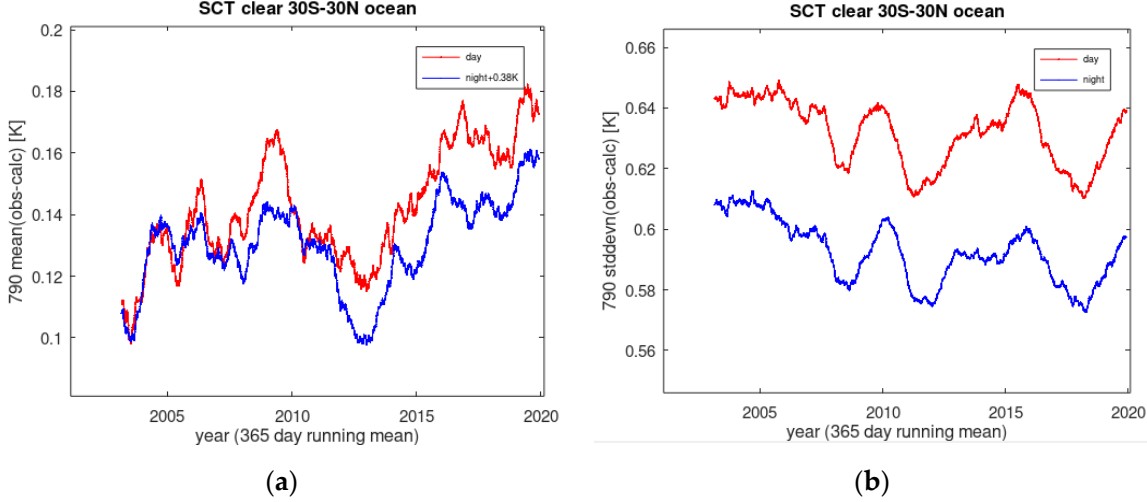

**Figure 5.** (**a**) Mean and (**b**) standard deviation at 790 cm$^{-1}$ with one-year smoothing.

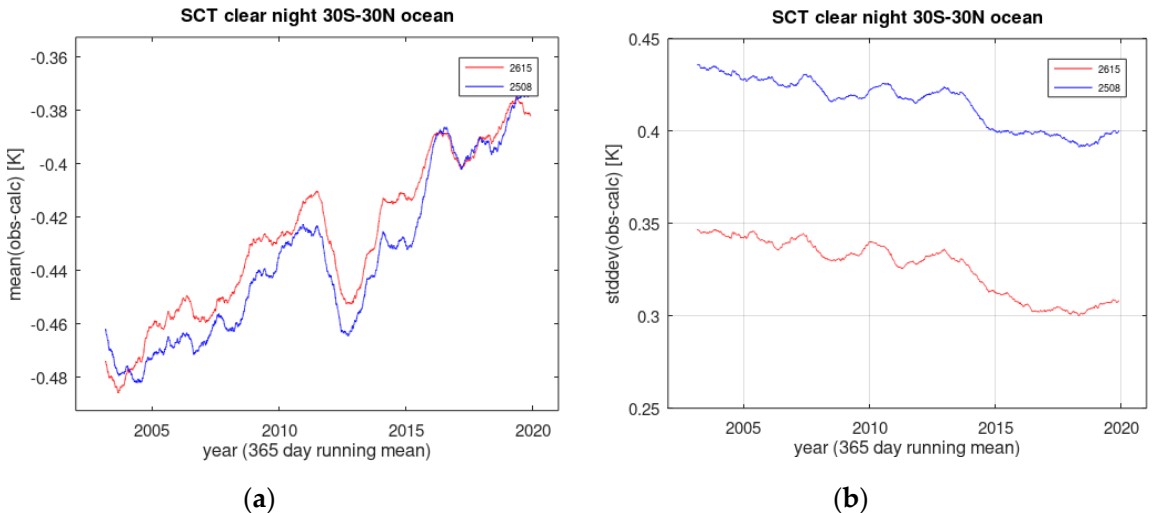

**Figure 6.** (**a**) Mean and (**b**) standard deviation at 2615 and 2508 cm$^{-1}$ with one-year smoothing.

**Table 2.** Summary of results from night (a) and day (b) overpasses.

(a) Night

| Channel [cm$^{-1}$] | Night Bias [K] | Stddev [K] | SCT = 0 Bias [K] | SCT Slope | 0.38K Night Bias Corr. [K] | Trend± 1σ [mK/yr] |
|---|---|---|---|---|---|---|
| 2615.3 | −0.59 | 0.326 | −0.38 | −0.22 | −0 | +5.6 ± 0.2 |
| 2508.1 | −0.435 | 0.415 | −0.38 | −0.18 | −0 | +5.7 ± 0.2 |
| 1231.3 | −0.59 | 0.405 | −0.49 | −0.22 | −0.11 | −2.2 ± 0.1 |
| 1128.5 | −0.404 | 0.486 | −0.32 | −0.20 | +0.18 | −0.4 ± 0.2 |
| 961.4 | −0.351 | 0.405 | −0.22 | −0.27 | +0.11 | −0.6 ± 0.1 |
| 901.0 | −0.333 | 0.472 | −0.21 | −0.27 | +0.11 | +1.8 ± 0.2 |
| 790.3 | −0.250 | 0.593 | −0.14 | −0.26 | +0.12 | +1.4 ± 0.2 |

**Table 2.** *Cont.*

(b) Day

| Channel [cm$^{-1}$] | Day Bias [K] | stddev | SCT = 0 Bias [K] | SCT Slope | Day Bias Corrected | Trend± 1σ [mK/yr] |
|---|---|---|---|---|---|---|
| 2615.3 | 3.83 | 4.14 | 3.82 | +0.13 | na | +10.3 ± 1.4 |
| 2508.2 | 1.92 | 2.18 | 1.96 | −0.02 | na | +9.1 ± 0.7 |
| 1231.3 | −0.22 | 0.451 | −0.15 | −0.15 | −0.15 | −0.6 ± 0.2 |
| 1128.5 | −0.02 | 0.516 | +0.05 | −0.13 | +0.05 | +1.7 ± 0.2 |
| 961.4 | +0.02 | 0.438 | +0.11 | −0.20 | +0.11 | +1.2 ± 0.2 |
| 901.0 | +0.04 | 0.514 | +0.19 | −0.19 | +0.19 | +3.8 ± 0.2 |
| 790.3 | +0.14 | 0.633 | +0.24 | −0.18 | +0.24 | +2.7 ± 0.2 |

## 4. Discussion

We discuss our results in terms of the bias, standard deviation, and anomaly trend of (obs-calc).

### 4.1. The Bias in (Obs-Calc)

Fiedler et al. [9] used 7 years of matchups to find the mean CMC minus Argo floating buoys difference to be the −10 mK level. This is an order of magnitude smaller than the bias between AIRS and the CMC seen in Table 2. The major contributors to this bias are residual cloud contamination, corrections for the diurnal effect and skin effects, water vapor correction uncertainty, and the absolute calibration of AIRS. Uncertainties in the absolute calibration of AIRS are dominated by uncertainties in the scan mirror polarization, the effective (as opposed to the telemetered) temperature of the on-board blackbody calibrator (OBC), and the pre-launch determined non-linearity coefficients. Errors due to non-linearity should be minimal, because the typical brightness temperatures of the observations are close to the OBC temperature. The bias in the shortwave channels (2615 cm$^{-1}$ and 2508 cm$^{-1}$) is higher than what is expected based on those uncertainties. There are indications of contamination of the scan mirror, causing an increase in scattering at these frequencies. This degradation could work its way into the radiometric calibration through either the OBC view, Space view, or the Earth view (via contribution from neighboring pixels) [12], resulting in the higher than expected bias relative to the CMC.

### 4.1.1. Cloud Contamination.

If 1% of a 300 K ocean footprint were to be covered by a 220K cloud, the effective mean temperature of the footprint at 1231 cm$^{-1}$ would decrease by 0.5 K. With a SCT clear threshold of 0.5 K, this footprint and an adjacent totally clear footprint would be identified as SCT clear. The presence of some clouds in some footprints which are identified as SCT clear will thus create a cold bias. The magnitude of the cloud contamination can be estimated by changing the threshold of the SCT filter. Figure 7 shows the decrease in the bias as the SCT threshold is changed from 2K to 0.5K. A tighter threshold results in a steep decrease in the yield as well. The SCT = 0.5 K threshold represents a practical limit. Extrapolated to SCT = 0 (as shown in Figure 7), the AIRS 1231 cm$^{-1}$ bias for day and night becomes −0.14 K and −0.49 K, respectively. The SCT = 0 extrapolated biases for all channels are listed in Table 2.

### 4.1.2. Diurnal and Skin Effects

The CMC represent the daily mean temperature at the buoy level, while SST.airs is the surface skin temperature at the time of the overpass. The skin is on average 0.2 K colder than the buoys [6], with a very small wind speed sensitivity [13]. Seasonally averaged between 20S and 20N, the buoy temperatures are 0.20 K warmer than the mean for the 1:30 PM overpasses and 0.11 K colder than the mean for the



1:30 AM overpasses [14]. Therefore, we expected SST.airs −CMCSST to be −0.2 K −0.11 K = −0.31 K at night, −0.20 K + 0.2 K = 0 K during the day. The observed (SCT = 0 extrapolated) day/night bias difference is 0.35 K. This 40 mK difference could be due to applying the buoy-mean difference from 20S–20N to the 30S–30N oceans.

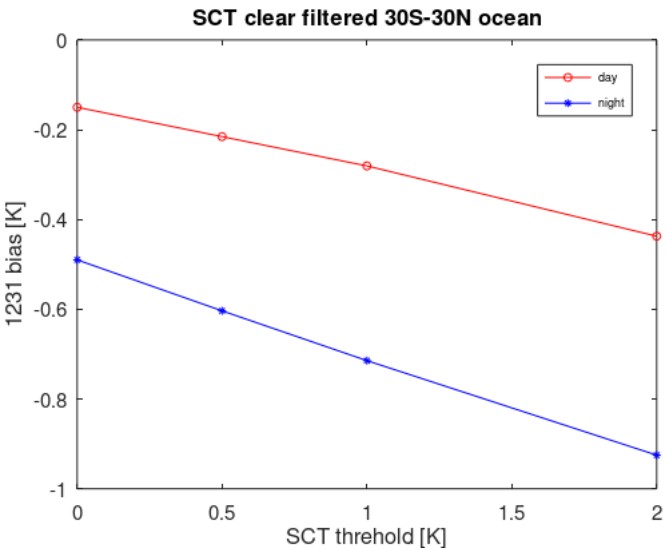

**Figure 7.** Cold bias as a function of the spatial coherence threshold (SCT).

### 4.1.3. Bias (Obs-Calc) Residuals

In the context of evaluating the absolute calibration, it is legitimate to correct the bias for cloud contamination and the diurnal effect. Table 2, column 6, lists the diurnal cycle and cloud contamination corrected bias. The residual bias ranges from −150 to + 240 mK.

The residual bias can at best only partially be attributed to an uncertainty of the transmission correction. The transmission correction ranges from 1K at 2616 cm$^{-1}$ to 5K at 790 cm$^{-1}$. The transmission correction was based on the water continuum used by the RTM and the vertical water vapor distribution ECMWF profiles in the training set. These profiles were a mix of convective (more wet) and subsidence (more dry) case, but the transmission correction was applied to clear sky (subsidence) cases. If the transmission correction were too weak by 1%, the correction would be between 10 and 50 mK too small, depending on the channel. This is too little to shift the −150 mK residual bias to zero, and would make the +240 mK bias even larger. In addition, the corrected residuals show only a weak correlation with the magnitude of the transmission correction. This suggests that the residual bias is dominated by absolute calibration effects. The SI traceable absolute calibration uncertainty for the AIRS detector modules in this study (Table 1, 3rd column), are reasonably consistent with the estimated bias residuals (Table 2, column 6). The comparison with the CMC does not significantly improve the absolute calibration uncertainty estimates.

### 4.2. The Standard Deviation (stddev(Obs-Calc))

The time series of the daily stddev(SST.airs-CMC) (Figures 1–6, right panels) show a different temporal structure than the time series of the daily mean. This structure is likely related to changes in the CMC. The stddev decreases steadily in all channels, day and night. Since the AIRS instrument or calibration did not change, the decrease in the stddev indicates a steady improvement in the fidelity of the CMC, most likely related to an increasing skill in the ingest of more satellite data. At 2615, 1231 and 961 cm$^{-1}$ the stddev is less than 0.4 K for the night data. This indicates a level of agreement between the CMC and our independent SST observations seen previously only relative to the Argo buoys in the 2000–2011 period [9].

The stddev for the day observations of the midwave channels is about 15% higher than that of the night observations at 1231 cm$^{-1}$. The most likely explanation for this is that the higher daytime yield in clear is associated with higher cloud contamination and associated additional noise. If the SCT threshold is raised from 0.5 K to 1 K, the yield of "clear" increases by a factor of 3 during the day, the cold bias increases from −0.21 K to −0.28 K, and the stddev increases from 0.44 K to 0.55 K.

The day and night results for the 2508 and 2616 cm$^{-1}$ channels are included in Table 2 to illustrate an unexpected daytime effect. The SST.airs calculation for these channels did not account for solar reflected radiation. It can be seen that the solar reflected component is about 2 K at 2508 cm$^{-1}$, 4 K at 2616 cm$^{-1}$, and highly variable due to its sensitivity to wave angle, whitecaps and residual cloud contamination. This causes the stddev for the day observations with the shortwave channels to be almost an order of magnitude larger than at night.

### 4.3. Anomaly Trends in Mean(Obs-Calc)

The time series of the mean(obs-calc) has a seasonal component, which is removed in the time series of the anomaly of mean(obs-calc). The anomaly trends of the mean(obs-calc), referred to as just "trends", are not consistent for the different channels. The trends in the midwave channels are day/night inconsistent based on the 1σ error bars, with trends ranging from −2.2 to +3.4 mK/yr. If we were to ignore the error bars and treat the 10 trends from the midwave channels as 10 independent samples of a distribution, we would state the trend as +0.9 mK/yr with 1.2 mK/yr 2σ confidence. This does not tell if AIRS is warming, or an artifact in the CMC causes the CMC to get colder.

Inspection of Figures 1–5 show what appears to be a change in the state of the instrument after 2012. The change appears to have a different effect on different detector modules, and differs between day and night, This eliminates the CMC as a potential reason. This effect is currently under investigation by the AIRS calibration team. Also visible in the time series of the bias is a 40 mK dip in the 1231 cm$^{-1}$ channel, seen day and night and centered on 2013, which is seen even more clearly in the 2615 and 2508 cm$^{-1}$ channels (Figure 6). This pattern is common to all channels and suggests an artifact in the CMC. The same pattern is seen in the comparison of CrIS SNPP data and the CMC SST [15].

The character of the anomaly trend of (obs-calc) of the two shortwave channels (Figure 6) is very different from that of the midwave channels. We see a 5 mK/yr warming trend at night and almost twice that during the day. This trend was briefly interrupted in 2013 by a 40 mK dip mentioned in the last paragraph, but then the trend continues. We interpret this trend to be an artifact of the L1B V5 calibration which is currently under investigation by the AIRS calibration team [12].

The EOS Aqua spacecraft with AIRS was launched into its 1:30 PM orbit in 2002, also known as the A-train. It is expected to exit the A-train in January 2022 and slowly drift to an increasingly later ascending node and lower altitude. The AIRS L1B v5 calibration is SI traceable and has been unchanged since launch. With 18 years of data, artifacts at the 100 mK absolute level, and trends well below 10 mK/yr, become visible in the AIRS data and in the reference truth data. After the exit from the A-train the AIRS calibration team will analyze the available data and correct those artifacts which can be physically related to events, voltages or temperatures on the spacecraft. This provides the opportunity for final refinements of the AIRS L1B calibration.

The AIRS instrument was designed to measure climate change. This task was facilitated by the actively maintained 1:30 PM ascending node of the EOS AQUA spacecraft orbit. The AIRS design life was 5 years, but by 2022 AIRS will have provided a continuous 20 year data record for climate change studies, the longest continuous data record to date from any temperature sounder.

## 5. Conclusions

We compare the daily mean and standard deviation of the difference between SST derived from clear AIRS data and collocated CMC data from the tropical oceans. After correcting the mean for cloud contamination and diurnal effects, the remaining bias relative to the CMC at the 100 mK level is reasonably consistent with estimates of the AIRS absolute accuracy based on the uncertainty of the

pre-launch calibration. The anomaly time series of the bias has channel-dependent trends, but all well below the 10 mK/yr level required for climate change evaluations. The trends are in the 2 mK/yr range for the midwave channels, but +5 mK/yr for the shortwave channels. The trend in the shortwave channels is likely due to a scan mirror degradation. The time series of the standard deviation of the difference between the AIRS SST and the CMC dropped steadily to below 0.4 K in several AIRS window channels, a level previously only seen in the CMC relative to the Argo buoys.

**Author Contributions:** Conceptualization, H.H.A.; Methodology, H.H.A.; Software, H.H.A., E.M.M. and R.C.W.; writing–original draft, H.H.A.; writing–review and editing, H.H.A., S.E.B., E.M.M., T.S.P., R.C.W.; Project Administration, T.S.P. All authors have read and agreed to the published version of the manuscript.

**Funding:** This research was carried out at the Jet Propulsion Laboratory, California Institute of Technology, under contract with NASA.

**Acknowledgments:** Jorge Vasquez, JPL, suggested the use of the CMC SST. The daily AIRS Calibration Data Subset (ACDS) is available free of charge from https://disc.gsfc.nasa.gov/datasets/AIRXBCAL_005/summary? keywords=AIRXBCAL. The CMC is freely available from https://podaac.jpl.nasa.gov/dataset/CMC0.2deg-CMC-L4-GLOB-v2.0?ids=&values=&search=CMC.

**Conflicts of Interest:** The authors declare no conflict of interest.

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
