# Peer review of "Evaluating the Absolute Calibration Accuracy and Stability of AIRS Using the CMC SST"

_remotesensing, doi:10.3390/rs12172743_

Round 1

Reviewer 1 Report

This is another good paper by the AIRS team, in which it compares the AIRS observations with CMC SST.  The paper is well written and results presented clearly.  It is acceptable for public with minor corrections and clarifications, specifically:

1) Abstract, line 2: clear AIRS data.  I guess this refers to the clear sky AIRS data.  It would be easier to understand if it says "clear sky" instead of just "clear" AIRS.

2) Lines 107: "could works its way", should it be "could work its way"?

3) Line 166: "has seasonal a component", probably should be "has a seasonal component".

4) Line 171: "we would state the trend as +0.9 mK/yr".  Please elaborate a bit more.  What is this trend?  How to interpret it?  Does this mean the AIRS has a positive trend relative to CMC SST?  Or does this mean the SST is warming at _0.9 mK/yr.  It's not clear based on what's presented here.  Please clarify.

Author Response

Out response is in sticky notes in the attached PDF file.

Reviewer 2 Report

This paper compares the absolute calibration of the stability of the AIRS sensors with respect to the CMC SST over tropical oceans. The TOA AIRS measurements are estimated at the surface using ECMWF atmospheric profiles. After the authors account for cloud contamination in their clear-sky filtering and diurnal differences they find the bias within the AIRS sensor calibration uncertainty.  I recommend this paper to be published after the following minor comments are addressed.

General Comments:

Is this the first time the AIRS calibration has been referenced to SST? Or are there AIRS SST products? If so in the introduction this should be mentioned and how this approach either differs or is similar.

Have the CMCSST been used to validate other IR hyperspectral sounders? Is it the gold standard of the SST community?

Specific comments.

Line 13 not inconsistent, why is there a double negative here? I would choose “within the estimates”

Line 17 “standard deviation of the difference between the AIRS SST and the CMC SST shows the effects of version changes in the CMC SST” Are the version changes due to various AVHRR entering and leaving the record? Are any of the satellites hosting AVHRR sensors in nearly the same Aqua orbit? Do the authors think that the individual AVHRR sensors are related to the variation in the standard deviation in figures 1b through 6b?

Line 51 Aqua-AIRS

Line 71 Have the ECMWF water vapor profiles become more accurate overtime, thus causing larger uncertainties in the earlier part of the record in Figures 1 through 6?

Figures 2a, 3a, 4a, 5a Why is the day temperature bias less than the night in the earlier record and opposite in the latter part of the record?

Line 103, How would mirror polarization, blackbody temperature and emissivity, and nonlinearity manifest themselves in this study? Nonlinearity is really an issue for cold scenes. I would think that the black body temperature is continuously monitored and accounted for. Does mirror polarization have any impact on the Earth emitted radiance (not the visible component)?

Line 126. The day and night bouy temperature difference would be the diurnal SST variation. The CMCSST is the 24-hour average, which takes into account the diurnal SST variation. Now looking at table 2 “0.38K night bias corr.” why does the day/night AIRS SST temperature difference increase with longer wavelengths?

Line 139 Would not the humidity distribution within the atmosphere impact the atmospheric correction. The humidity in the upper atmosphere is more likely to reduce the radiance leaving the surface, than if the humidity was located closer to the surface. Does the transmission term take into account where the humidity is placed? The tropics contain both subsidence and convective areas. Maybe I am missing something.

Line 179 I believe it is appropriate to indicate that changes in polarization over time results in a large daytime trend of the shortwave channels in this paragraph, just as you state in the conclusions.

Author Response

The response is in sticky notes in the pdf file

Reviewer 3 Report

Evaluating the Absolute Calibration Accuracy and Stability of AIRS using the CMC SST

Hartmut H. Aumann1, Steve Broberg1, Evan Manning1, Tom Pagano1 and R.C. Wilson1

Submitted to Remote Sensing ref 895164 (letter)

General comments

This letter is focused on the quality of SST retrieved from data collected by the Atmospheric InfraRed Sounder (AIRS) data on board the AQUA spacecraft between 2002 and 2020. The main conclusion is that these data, when compared to the CMC SST, best multi sensor dataset so far, offer performances in terms of std which improve with time and become close to the performances of the CMC dataset when compared to the reference dataset supplied by the Argo buoys. Furthermore, the trend magnitude is shown to meet the requirements necessary to monitor adequately the impact of global warming.

These results deserve to be reported, as they illustrate and confirm significant successes of the hyperspectral instruments and specifically AIRS as well as its processing.

The letter is in general clearly written. Obviously, it will not however be easy reading for readers who are not somewhat acquainted with the relevant instrument and techniques, but after all references are provided and available.

Although the letter aims at pointing out achievements of the AIRS, it also mentions some pending anomalies which are under investigation. This is to be commended.

A small number of minor corrections is indicated below.

Detailed comments

L101: I suspect it is table 2 rather than table 1

L114: the comma following "clear" seems spurious

L130-131: I am ready to believe you; however, a short explanation or supplying a reference would do no harm.

L137: It is suggested to insert "(3rd column in Table 1)"

L147: concerning the 2615 cm^-1 data, they are displayed on figure 6

L166: order of words to be inverted.

Author Response

Our response to reviewers 1-4 is in the sticky notes in the pdf file.

Reviewer 4 Report

See attached comments.

Author Response

imbedded in the pdf of the manuscript
